# Optimization of Neferine Purification Based on Response Surface Methodology and Its Anti-Metastasis Mechanism on HepG2 Cells

**DOI:** 10.3390/molecules28135086

**Published:** 2023-06-29

**Authors:** Xinzhu Wang, Zhenhuan Wei, Po Hu, Weibo Xia, Zhixin Liao, Israa Assani, Guangming Yang, Yang Pan

**Affiliations:** 1School of Pharmacy, Nanjing University of Chinese Medicine, 138 Xianlin Avenue, Qixia District, Nanjing 210023, China; xzwang@njucm.edu.cn (X.W.); 20200860@njucm.edu.cn (Z.W.); hupo_cpu@foxmail.com (P.H.); xwb15366067163@163.com (W.X.); 2Department of Pharmaceutical Engineering, School of Chemistry and Chemical Engineering, Southeast University, Nanjing 211189, China; zxliao@seu.edu.cn (Z.L.); israaassani@hotmail.com (I.A.)

**Keywords:** Neferine, response surface methodology (RSM), HepG2, migration and invasion, RhoA

## Abstract

Liver cancer continues to be a focus of scientific research due to its low five-year survival rate. One of its main core issues is the high metastasis of cells, for which there is no effective treatment. Neferine was originally isolated from *Plumula nelumbinis* and demonstrated to have a good antitumor effect. In order to extract high-purity Neferine in a more efficient and environmentally friendly manner, response surface methodology (RSM) was used to optimize the isolation and purification procedures in this study. The extract conditions of a 7:3 ratio for the eluent of dichloromethane: methanol, 1:60 for the mass ratio of the extract amount: silica gel, and 3 mL/min of the elution flow rate were shown to be the optimal conditions. These conditions resulted in the highest yield of 6.13 mg per 66.60 mg of starting material, with productivity of 8.76% and purity of 87.04%. Compared with the previous methods, this method can prepare Neferine in large quantities more quickly. We subsequently evaluated the antitumor activity of the purified Neferine against HepG2 hepatic cancer cells. The purified Neferine was found to inhibit the proliferation of HepG2 cells through the CCK-8 assay, with an *IC*_50_ of 33.80 μM in 24 h, 29.47 μM in 48 h, 24.35 μM in 72 h and 2.78 μM in 96 h of treatment. Neferine at a concentration of 3 μM could significantly inhibit the migration and invasion abilities of the HepG2 cells in vitro. We also explored the mechanism of action of Neferine via Western blot. We showed that Neferine could reduce RhoA expression by effectively inhibiting the phosphorylation of MYPT1, thereby effectively exerting anti-metastasis activity against HepG2 cells. Thus, we have optimized the isolation procedures for highly pure Neferine by response surface methodology (RSM) in this study, and purified Neferine is shown to play an essential role in the anti-metastasis process of liver cancer cells. The Neferine purification procedure may make a wide contribution to the follow-up development of other anti-metastasis lead compounds.

## 1. Introduction

Liver cancer continues to be a worldwide health issue, with a high mortality rate [1,2,3,4] and a relatively low five-year survival rate of approximately 18% [5]. Liver cancer is a malignant tumor with no complete cure. Sorafenib (a kinase inhibitor) is a prescription drug for treating liver cancer. However, long-term use of sorafenib may lead to drug resistance and side effects [6]. Therefore, there is an urgent need to identify novel leading compounds and develop new therapeutic strategies to treat liver cancer with low or non-toxic side effects for patients.

Lotus (*Plumula nelumbinis*) is well-known for its use as medicine, and lotus seeds and roots are also used as food in daily life. As a traditional Chinese medicine, *Plumula nelumbinis* has been proven effective in clearing heart heat, calming the mind, and promoting astringent essence and hemostasis (Chinese Pharmacopoeia 2020). The main active substance of *Plumula nelumbinis* is identified as Neferine [7], which has a range of therapeutic effects, including lowering blood pressure [8], anti-platelet aggregation [9], anti-arrhythmia [10,11], and especially antitumor [12,13,14]. Although there are various methods for preparing Neferine [15,16,17,18,19], the existing methods have some deficiencies and need further optimization and improvement. Response surface methodology (RSM) can be used to evaluate the effect of processing parameters on response value by planning experiments and building models. RSM has previously been used to optimize the extraction process of bioactive compounds [20], but it has not yet been examined if it is a reliable choice for isolating Neferine.

Neferine has a good therapeutic effect on liver tumors, but the mechanism of action is not entirely clear. In this research, we improved the Neferine isolation process to make it more efficient and environmentally friendly. Meanwhile, we demonstrated the anti-proliferative activity of the purified Neferine on HepG2 cells via the CCK-8 assay and inhibitory effects on the invasion and migration of HepG2 cells through transwell and cell scratch experimentation. We explored the mechanism of its antitumor effect and showed that Neferine could regulate the RhoA/Rho pathway.

## 2. Results

### 2.1. The Standard Curve of Neferine

The Neferine standard was made by dissolving 5.03 mg of purified Neferine in 5 mL of methanol. Then, 1, 1.5, 2, 2.5, and 3 mL of the reference solution were accurately prepared and injected into HPLC for measurement. The standard curve (plot) was plotted with the peak area as the ordinate (Y) and the control concentration as the abscissa (X), and the linear regression equation was calculated as y = 4877.1x + 270.44, R^2^ = 0.996 (Figure 1). These results indicate that this standard curve is reliable as a reference.

### 2.2. Model Establishment and Significance Analysis

The simulated and actual outputs are shown in Table 1 through 17 random combinations of flow matching ratio (mL/mL), extract amount: silica gel amount (g:g), and elution flow rate (mL/min). As mentioned above, the purity of Neferine obtained in the actual experiment was close to the predicted value, indicating that the model was well-fitting and highly accurate.

Out of 17 experimental conditions tested, the combination of a 7:3 eluent ratio of dichloro-methane to methanol, a mass ratio of 1:60 for the extract amount to silica gel, and an elution flow rate of 3 mL/min showed the highest yield. This combination led to the highest yield of 6.13 mg per 66.60 mg of starting material, with a productivity of 8.76% and a purity of 87.04%. This combination was superior to others, as demonstrated by the variance analysis, where coefficient B^2^ was significant (*p* < 0.01), indicating that the variables B^2^ could significantly influence the extraction yield of Neferine. The same was true for productivity, with *p* < 0.01. For the purity, the variance analysis demonstrated that coefficients C and A^2^ were significant (*p* < 0.01), which indicated that the variables C and A^2^ could significantly influence the extraction purity of Neferine. The *p* < 0.0001 of this mode and the respective *p*-values of 0.064, 0.064, and 0.058 for the lack-of-fit value indicated insignificant deviation effects, suggesting a fitted model (Table 2).

### 2.3. Interaction Effect of Each of the Two Parameters

Figure 2 illustrates the interaction effects of the eluent polarity (9:1, 7:3, 6:4, and 5:5, marked as A), the proportion of extract volume to silica gel volume (1:20, 1:40, 1:60, and 1:80, marked as B), and the elution flow rate (1, 2, 3, and 4 mL/min, marked as C) on the purity, productivity, and yield of Neferine. The influence of these two interactive factors on the response value was more significant when the contour was closer to the ellipse or when the response surface curve was steeper. Conversely, the influence was deemed insignificant when the contour was further from the ellipse or the response surface curve was less steep. Therefore, the polarity of eluent and elution flow rate (A and C), and the proportion of extract volume to silica gel volume (A and B) exhibited significant effects (*p* < 0.01) on the yield, productivity, and purity of Neferine, which were consistent with previous results.

The yield, productivity, and purity were referred to as R1, R2, and R3, as shown in Figure 2. For yield (R1) and productivity (R2) of Neferine, the ratio of extract volume to silica gel volume, the proportion of mobile phase, the elution flow rate and flow matching ratio, and the ratio of elution flow rate and extract volume to silica gel volume all displayed statistically significant differences. For purity (R3), the ratio of the amount of extract to the amount of silica gel and the matching flow ratio showed a statistically significant impact. In contrast, the elution flow rate and the flow matching ratio—the elution flow rate or the ratio of the amount of extract to the amount of silica gel—exhibited no significant influence on the purity (R3) of Neferine.

The optimal extraction conditions were determined by Design–Expert.V8.0.6.1 software. The optimal conditions for the separation and purification by silica gel chromatography were set as follows: the eluent (dichloromethane:methanol) ratio was 6.83:3.17; the mass ratio of extract amount to silica gel amount was 1:57; and the elution flow rate was 2.86 mL/min. To facilitate practical experimentation, we defined the best processing conditions as follows: 7:3 for dichloromethane: methanol; 1:60 for the mass ratio of extract amount: silica gel; and 3 mL/min for the elution flow rate. In this experiment, the separation process of Neferine was optimized by the orthogonal method. The optimal process was finally obtained from the response surface method based on the yield and purity of Neferine at 282 nm. The refined extraction method could help improve the yield and purity of Neferine. Meanwhile, it could be safer for the environment, significantly reduce the time of separation and purification, and provide a material basis for further bioactivity determination.

### 2.4. Preliminary Analysis of Neferine

The obtained fractions were detected by HPLC and compared with the standard in terms of peak time and UV absorption wavelength. Its molecular weight was determined by LC-MS (Appendix A), and its structure was revealed by NMR (Appendix A) [21]. These data together showed that the obtained fraction was Neferine.

### 2.5. Neferine Inhibits the Proliferation of HepG2 Cells

The cytotoxicity of Neferine was subsequently determined in HepG2 cells by the CCK-8 assay. Cells were treated with different concentrations of Neferine (16, 32, 48, 64, and 80 µM) for 24, 48, 72, and 96 h (Figure 3). The *IC*_50_ was shown to be 33.80, 29.47, 24.35, and 2.78 μM, respectively. These data suggested that Neferine could suppress the proliferation of HepG2 cells at a micromolar concentration after four days of treatment.

### 2.6. Neferine Inhibits the Migration of HepG2 Cells

The cell scratch experiment was carried out, and the scratched space was measured during the subsequent culture for wound healing. The data showed that Neferine induced a marked decrease in cell migration of HepG2 cells in a dose-dependent manner (*p* < 0.05 and *p* < 0.001; Figure 4b). The transwell assay also revealed that Neferine significantly inhibited the migration and invasion of HepG2 cells in a dose-dependent manner (*p* < 0.0001; Figure 4d,f).

### 2.7. Changes of Related Protein Expression Induced by Different Dosage of Neferine

Different concentrations of Neferine were used to treat cells, and the proteins were extracted after 24 h. The expression of Vimentin was significantly decreased (Figure 5c) after Neferine treatment, consistent with inhibited effects and reduced cell migration. The expression of RhoA was also decreased significantly (Figure 5b), but at the concentration of 3 μM, the expression of RhoA protein abundance was not significantly different from the control group. This was not surprising because 3 μM was close to the *IC*_50_ of 96 h treatment, which indicated that a low dose of Neferine had little effect on the expression of RhoA. Indeed, when the dose was increased to 6, 12, or 16 μM, Neferine could significantly reduce the expression of RhoA, indicating that Neferine could inhibit the expression and/or stability activity of RhoA. For p-MYPT1, even 3 μM Neferine at even 3 μM could significantly inhibit its expression (Figure 5d), indicating that p-MYPT1 was highly sensitive to Neferine treatment and that Neferine had a very high inhibition effect on p-MYPT1, which could be inhibited considerably at a very low dose.

## 3. Discussion

In the current research, countercurrent chromatography and preparative high-performance liquid chromatography extraction methods are relatively novel [22,23,24,25]. This method can separate Neferine with high purity, but the procedures come at the cost of complex dissolution selection [26], a longer separation time [27], and higher pH requirements [28]. However, preparing the liquid phase requires a small amount of sample loading, a long separation cycle, low separation efficiency per unit time [29], and a large amount of solvent. The above two methods do not align with our demand for obtaining Neferine more quickly. Medium-pressure silica gel column chromatography may result in dead adsorption and the loss of required components. Still, it can process a large number of samples and obtain the relatively objective quality of Neferine within a few hours while ensuring purity. Compared with the other two methods, medium-pressure silica gel chromatography has a relatively shorter time and lower economic cost, making it more suitable for various types of laboratory work. Compared to other methods, the separation efficiency per unit time of medium-pressure silica gel separation is greatly increased, and the separation cost is relatively reduced. Therefore, medium-pressure silica gel column chromatography is more in line with our experimental requirements for obtaining Neferine in large amounts but in a short time.

Hepatocellular carcinoma (HCC) is one of the most common cancers in the world due to its low five-year survival rate and high mortality rate, which brings a huge burden to the global medical system [5]. In molecular pathogenesis, dysregulation of key signaling pathways, i.e., Wnt/β-catenin, JAK/STAT, etc., strongly promotes the development of HCC [30,31,32,33]. Although significant progress has been made in the diagnosis and treatment of liver cancer, including radiotherapy [34], chemotherapy [35], and immunotherapy [36], and first-line clinical drugs have been introduced to treat advanced liver cancer [37,38], the prognosis of some HCC patients is not ideal, and their five-year survival rate is no more than 18% [5]. One of the most important factors is the metastasis of liver tumor cells [39,40].

The sustained growth of cancer cells is due to the presence of continuous value-added signals, which leads to the rapid growth of cancer cells [41]. The poptosis of cells is mediated by cascading signals of cell apoptosis [42]. Cancer cells can escape this committed step and thus extend their life span [43]. Neferine can trigger cell apoptosis through various pathways, as Zhao et al. show that Neferine has an inhibitory effect on CYP1A2, CYP2D6, and CYP3A4 in vitro [44]. Paramissivan Poornima et al. also found that the mitochondrial membrane potential of HepG2 cells induced by Neferine can be decreased, the level of reactive oxygen species may be increased, and the expression of related apoptosis proteins such as Bax and caspase-3 can be upregulated. Neferine has an inhibitory effect on HepG2 cells in a dose-dependent manner [45]. Jin Soo Yoon et al. also concluded in the HepB3 cell line that Neferine is not toxic to normal hepatocytes. Still, it can induce cell cycle arrest and apoptosis by activating death receptors and mitochondrial-dependent pathways related to Endoplasmic reticulum stress and autophagy [46].

Tumor metastasis is the cause of over 90% of cancer deaths [47,48]. Deng et al. show that Neferine can affect snail protein, make it overexpressed, inhibit the EMT process of HepG2 cells, and enhance the sensitivity of Oxaliplatin to liver cancer [49]. The above studies indicate that Neferine has good antitumor activity and an inhibitory effect on the metastasis of liver tumor cells. However, its mechanism of action is not yet fully understood. The RhoA/Rho pathway is one of the important pathways implicated in tumor cell migration and invasion [50]. We previously found that Neferine inhibited the migration and invasion of non-small cell lung cancer through the RhoA/Rho pathway [51]. The characteristic of multiple components and targets in traditional Chinese medicine is well accepted by the public. Therefore, we explored whether Neferine inhibited the migration and invasion of HepG2 cells through the related pathway. In this study, the RhoA/Rho pathway was taken as the research object, and the upstream RhoA target and the downstream p-MYPT1 target were investigated. We provided the preliminary data for the mechanistic approaches of Neferine and proposed RhoA/Rho as one of its anti-HepG2 cell migration invasion pathways for multi-target therapy of liver cancer. The reorganization of the actin cytoskeleton and regulation of the morphology, attachment, and movement of cells were promoted by the Rho protein. The proliferation and metastasis of tumor cells were closely related to the overexpression of the Rho protein. RhoA protein is highly expressed in gastric cancer [52,53], breast cancer [54], colon cancer [55], pancreatic cancer [56], lung cancer [57], and other cancers. The previous study of liver cancer showed that overexpression of the RhoA protein could facilitate the migration and invasion of liver cancer cells [58,59]. The previous research aroused our great interest. Therefore, we conducted the anti-migration experiment on liver cancer cells targeting RhoA and explored its mechanism.

RhoA is upstream of MYPT1, and the high expression of RhoA leads to the decline of MYPT1 expression [60]. High expression of MYPT1 is shown to inhibit the proliferation and migration of cancer cells [61,62,63]. There is a negative regulatory relationship between the MYPT1 protein and RhoA [60]. The increased expression of the MYPT1 protein means that the phosphorylation level of the protein is reduced. This also means that the less p-MYPT1, the more abundant MYPT1. Therefore, this may infer that p-MYPT1 has a positive regulatory effect on RhoA. That is, a low level of p-MYPT1 can decrease the expression of RhoA, thus inhibiting the migration and invasion of tumor cells.

Based on the above theory, we studied one of the main active ingredients of *Plumula nelumbin*, Neferine. Through response surface methodology, the isolation process of Neferine on the medium-pressure silica gel column was optimized, enabling us to obtain Neferine more efficiently and environmentally friendly with higher purity, laying a solid material foundation for the subsequent activity experiment. Neferine has been proven to have a strong inhibitory effect on HepG2 cells. It could significantly inhibit the proliferation of HepG2 cells at micromolar concentrations (Figure 3). The results of the cell scratch assay and transwell test showed that Neferine could significantly inhibit the migration and invasion of HepG2 cells at a micromolar concentration level (Figure 4). Through the study of related pathway proteins, it was found that Neferine at 3 μM could significantly decrease the expression of Vimentin (*p* < 0.01). Neferine at 6 μM could significantly inhibit the expression of RhoA (*p* < 0.05), and at an increasing dose, the inhibitory effect also increased. Surprisingly, compared with the control group, p-MYPT1 was not expressed and was strongly inhibited after 24 h of stimulation with 3 μM Neferine. Therefore, we can speculate that Neferine can inhibit the migration and invasion of HepG2 cells by inhibiting the phosphorylation of MYPT1 and reducing the expression of RhoA (Figure 6). Compared with some traditional Chinese medicine monomers, such as berberine [64] and lycorine [65], Neferine shows a more prominent antitumor metastasis effect at a lower dose.

## 4. Materials and Methods

### 4.1. Materials

Experimental medicinal materials were purchased from Fuzhou, Jiangxi Province. Methanol from CR, TEDIA, Ltd. (Fairfield, OH, USA) and AR, Wuxi Yasheng Chemical Co., Ltd. (Wuxi, China); dichloromethane from AR, Wuxi Yasheng Chemical Co., Ltd.; and dimethyl sulfoxide from AR, China National Pharmaceutical Group Corporation, Ltd. (Hong Kong, China); C18 chromatographic column from Agilent (Santa Clara, CA, USA); medium-pressure silicone column from Grace (Houston, TX, USA); and ammonium acetate from AR, China National Pharmaceutical Group Corporation, Ltd. (Beijing, China).

Embryos of lotus seed were purchased from Fuzhou Medical Material Market (Fuzhou, China), collected from Jiangxi province, and authenticated by Professor Yang Pan (Nanjing University of Chinese Medicine, Nanjing, China) according to the Chinese Pharmacopoeia 2020.

### 4.2. Preparation Process of Neferine

#### 4.2.1. Preparation Process of Fatty Soluble Alkaloids from *Plumula nelumbinis*

A total of 70% alcohol was used to soak and extract *Plumula nelumbinis*, then the solvent was recovered, and 10% sulfuric acid was added to adjust the pH of the leftover to about 2–3. The residual solution was filtered and reduced by dichloromethane. The pH of the remains was modified to about 9–10 with ammonia. Dichloromethane was used to extract the solution that was obtained in the previous step, and the fatty soluble alkaloids were obtained.

#### 4.2.2. Isolation and Purification of Neferine

A medium-pressure silica gel column separation system was used at 26 °C to isolate Neferine with dichloromethane (A) and methanol (B) as the eluents. The crude extracts of lotus alkaloids were weighed according to the required loading ratio, then dissolved in dichloromethane and wet loaded into the medium-pressure preparation column. Gradient elution was carried out from 100% dichloromethane (A) to 100% methanol (B). All fractions were collected, and eight fractions were selected from dichloromethane: methanol (9:1), (7:3), (6:4), (5:5), and (0:10) for HPLC detection. The fractions were collected, dried, and dissolved in 1 mL of methanol. An Agilent 1260 HPLC was used to test the purity of previously isolated fractions. The result was compared with the AccuStandard of Neferine, and software (Agilent 1260 offline software) was used to calculate the purity, productivity, and yield.

### 4.3. Single-Factor Experiments

The effect of three factors on extraction efficiency was evaluated, including the polarity of the eluent (9:1, 7:3, 6:4, 5:5), the ratio of the extract amount to the silica gel amount (1:20, 1:40, 1:60, 1:80), and the elution flow rate (1, 2, 3, 4 mL/min).

#### RSM Modeling and Experimental Design

The preparation process of Neferine was improved by RSM’s Box Behnken Design (BBD). Three operating parameters, including the polarity of eluent (A), the proportion of extract volume to silica gel volume (B), and the elution flow rate (C), were encoded at the following three levels: −1, 0, and +1, respectively (Table 3).

### 4.4. Cell Culture

The HepG2 hepatoma cell line was purchased from Wuhan Servicebio Company. Cells were maintained within a humidified incubator with 5% CO_2_ at 37 °C and cultured with Dulbecco’s modified Eagle’s medium (DMEM, Gibco, Grand Island, NY, USA) supplemented with 10% fetal bovine serum (FBS, Biosharp, Tallinn, Estonia). The medium was replaced every 48 h. When the cell density reached 80% or above, trypsin (Beyotime Biotechnology, Shanghai, China) was used for digestion and passage.

### 4.5. CCK-8 Assay

Cells were treated with various concentrations of Neferine (16, 32, 48, 64, or 80 µM). Then, 100 µL of CCK-8 solution (10% CCK-8 and 90% DMEM) was added to each well after 24 h, 48 h, 72 h, or 96 h of culture, followed by an additional 2–4 h of incubation at 37 °C in a cell culture chamber. After that, the optical density at 450 nm was recorded by a microplate reader.

### 4.6. Wound Healing Assay

HepG2 cells were seeded in six-well plates at an appropriate density. After incubation, the cells in the 6-well plate reached about 90% confluence. A wound was generated by scratching with a 10 µL pipette and then washed with PBS to remove residual cells from the scratches. Neferine at 3, 6, 12, or 16 µM was added to the scratched culture and incubated for 24 h. The culture medium was replaced, and the cells were imaged. Additionally, the width of the gaps was quantified by ImageJ software.

### 4.7. Transwell Assay

HepG2 cells were adjusted to a density of 2 × 10^4^ cells/well. Culture media containing different FBS concentrations were added in the upper and lower chambers. Neferine at 3, 6, 12, and 16 µM was added to the two chambers and incubated for 24 h. Cells were fixed with 4% paraformaldehyde and stained with 0.1% crystal violet. A Leica microscope was used to image the migrated cells. The number of cells was calculated from three different groups and three independent experiments.

### 4.8. Western Blot Analysis

HepG2 cells were grown at 0 (control group), 3, 6, 12, and 16 µM of Neferine for 24 h. Cells were washed in 1×PBS and then RIPA lysis was used to extract total cell protein. Proteins were resolved on SDS-PAGE and transferred to the PVDF membrane. The blots were blocked at room temperature and incubated with the appropriate primary antibody at 4 °C overnight. The membranes were developed using the ECL system. β-actin was used as an internal control to ensure the equal loading of proteins. Finally, ImageJ software was used for the data analysis of stripes.

## 5. Conclusions

In our experiment, the isolation and purification of Neferine, the main active compound of *Plumula nelumbin*, was optimized by RSM to obtain a more efficient and environmentally friendly separation process with medium-pressure silica gel. Its anti-migration and invasion on liver tumor cell HepG2 were studied. It was found that Neferine could effectively inhibit the proliferation of HepG2 cells and then inhibit the RhoA/Rho pathway by repressing the expression of p-MYPT1. Neferine plays an important role in the anti-migration and invasion of liver tumor cells. Our experiment provides a scientific and experimental basis for the effective components of natural drugs to inhibit liver tumor metastasis. However, the metastasis of liver cancer involves many pathways, and the pathogenesis is relatively complex. Our experiment only focused on the RhoA/Rho pathway and carried out a series of studies. Whether Neferine is equally effective for other metastasis-related pathways was worth investigating in the follow-up studies but was beyond the scope of the current exploration. The search for pharmacologically effective components from plants used as food and medicine is receiving increasing attention from the scientific community. This study provides a scientific reference for the efficient development and utilization of traditional Chinese medicine resources. It offers a scientific and experimental basis for identifying effective components fighting against liver tumor metastasis from the same plants used as food and medication.

## Figures and Tables

**Figure 1 molecules-28-05086-f001:**
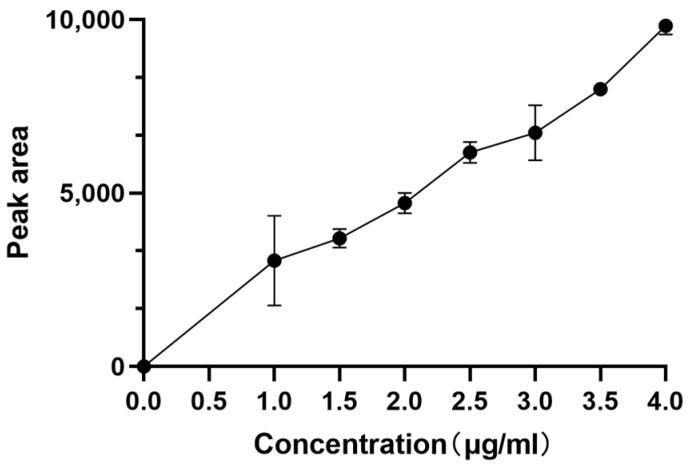
Standard curve of Neferine.

**Figure 2 molecules-28-05086-f002:**
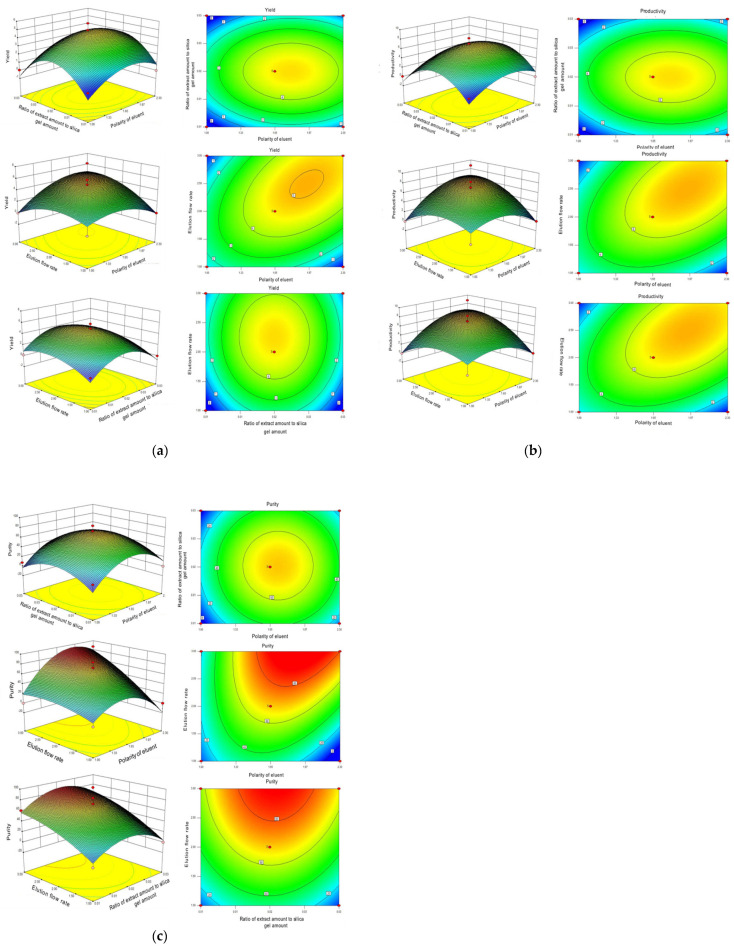
Optimization of the purification process of Neferine by response surface methodology. (**a**) Contour diagram of separation purity; (**b**) contour diagram of separation yield; and (**c**) contour diagram of separation yield.

**Figure 3 molecules-28-05086-f003:**
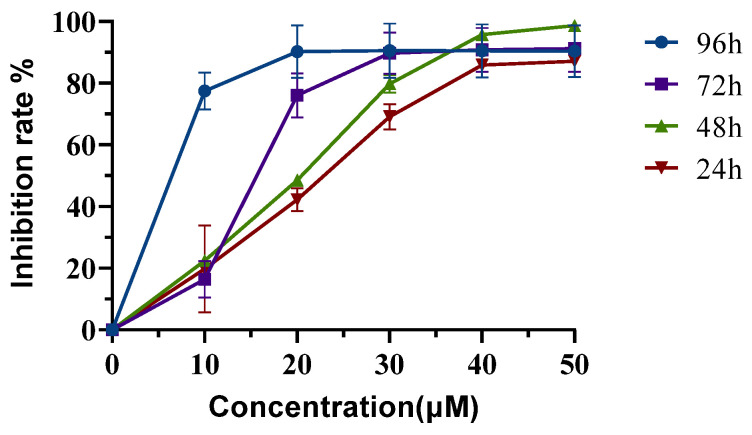
Neferine inhibits the proliferation of HepG2 cells. HepG2 cells were treated without or with Neferine for 24, 48, 72, or 96 h at defined concentrations. The *IC*_50_ was calculated from the inhibition rates during different lengths of treatment.

**Figure 4 molecules-28-05086-f004:**
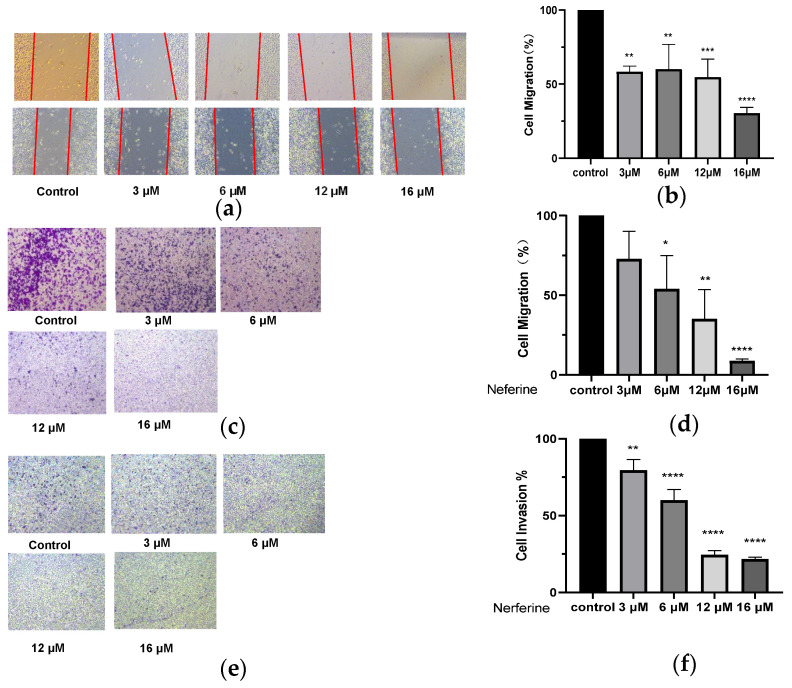
Effects of Neferine on migration and invasion of HepG2 cells. (**a**) After scratching the cells, Neferine at 3, 6, 12, or 16 μM was added for processing for 24 h of treatment. (**b**) Quantitative analyses of the scratch experiments. (**c**) Cells were treated with Neferine (3, 6, 12, or 16 μM) for 24 h, and (**d**) the effect of Neferine on the migration of HepG2 cells was assessed in a Transwell assay. (**e**) Cells were treated with Neferine at 3, 6, 12, or 16 μM for 24 h, and (**f**) the effect of Neferine on the invasion of HepG2 cells was assessed in a Transwell assay. * *p* < 0.05, ** *p* < 0.01, *** *p* < 0.001, **** *p* < 0.0001 vs. Control 4.

**Figure 5 molecules-28-05086-f005:**
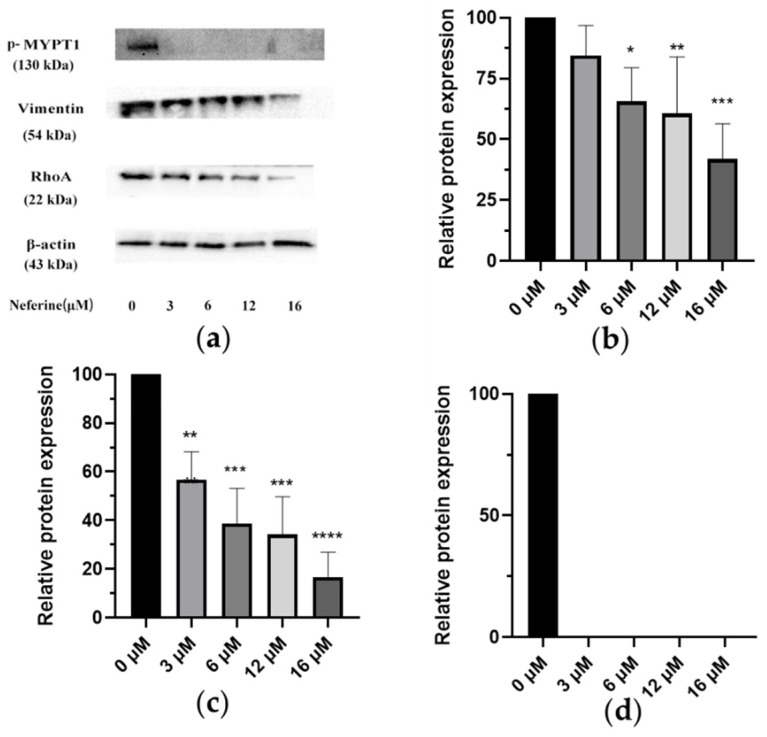
Changes in related protein expression induced by different concentrations of Neferine. (**a**) HepG2 cells were treated with Neferine (0, 3, 6, 12, and 16 μM) for 24 h, and a Western blot assay was performed to detect the levels of p-MYPT1 (130 kDa), RhoA (22 kDa), and Vimentin (54 kDa). (**b**) Effect of Neferine on RhoA. (**c**) Effect of Neferine on Vimentin. (**d**) Effect of Neferine on p- MYPT1. * *p* < 0.05, ** *p* < 0.01, *** *p* < 0.001, **** *p* < 0.0001 vs. control.

**Figure 6 molecules-28-05086-f006:**
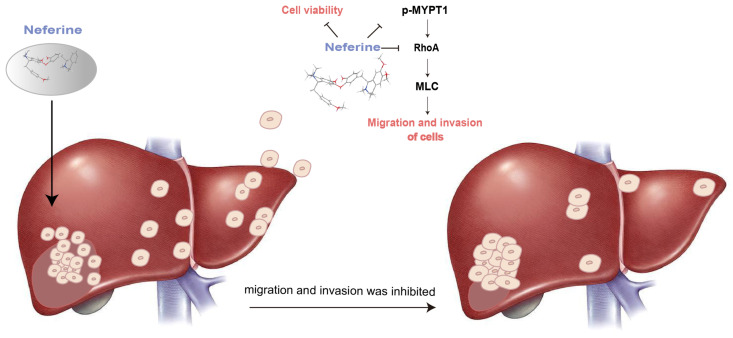
Mechanism of Neferine in inhibiting the migration and invasion of hepatoma cells.

**Table 1 molecules-28-05086-t001:** Box–Behnken response surface optimization of the design and results of the purification process of Neferine by silica gel column chromatography.

No.	Flow Matching Ratio (mL/mL)	Extract Amount: Silica Gel Amount (g: g)	Elution Flow Rate (mL/min)	Yield of Neferine (mg)	Productivity of Neferine (%)	Purity of Neferine (%)
1	7:3	1:40	2	0	0	0
2	6:4	1:60	2	5.76	8.23	58.32
3	7:3	1:60	1	0	0	0
4	7:3	1:80	2	0	0	0
5	5:5	1:60	3	0	0	0
6	6:4	1:40	3	0.19	0.19	73.68
7	6:4	1:80	3	0.17	0.34	60.49
8	5:5	1:40	2	0.05	0.05	7.79
9	6:4	1:60	2	4.49	6.42	72.95
10	6:4	1:40	1	0	0	0
11	7:3	1:60	3	6.13	8.76	87.04
12	5:5	1:60	1	0	0	0
13	6:4	1:80	1	0	0	0
14	5:5	1:80	2	0.05	0.10	9.22
15	6:4	1:60	2	4.96	7.09	64.14
16	6:4	1:60	2	4.51	6.44	67.87
17	6:4	1:60	2	3.78	5.41	83.65

**Table 2 molecules-28-05086-t002:** Quadratic model by ANOVA.

	Sum of		Mean	F	*p*-Value	
Source	Squares	df	Square	Value	Prob > F	
Model	19,082.55	9	2120.28	7.325	0.008	significant
A	612.87	1	612.87	2.117	0.189	
B	17.27	1	17.26	0.060	0.814	
C	6116.56	1	6116.56	21.131	0.003	significant
AB	0.51	1	0.51	0.002	0.968	
AC	1893.96	1	1893.92	6.543	0.038	significant
BC	43.47	1	43.47	0.150	0.710	
A^2^	6227.51	1	6227.51	21.514	0.002	significant
B^2^	2996.16	1	2996.16	10.351	0.015	significant
C^2^	354.09	1	354.09	1.223	0.305	
Residual	2026.22	7	289.46			
Lack of Fit	1657.83	3	552.61	6.000	0.058	not significant
Pure Error	368.40	4	92.10			
Cor Total	21,108.77	16				

**Table 3 molecules-28-05086-t003:** Optimization of factor level design for the purification process of Neferine by silica gel column chromatography with response surface methodology.

Level	A (mL/mL)	B (g:g)	C (mL/min)
−1	5:5	1:80	1
0	6:4	1:60	2
1	7:3	1:40	3

## Data Availability

All data generated or analyzed during this study are included in this published article (and its Appendix A).

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
