# Peer review of "Optimization of Neferine Purification Based on Response Surface Methodology and Its Anti-Metastasis Mechanism on HepG2 Cells"

_molecules, 2023, doi:10.3390/molecules28135086_

Round 1
Reviewer 1 Report
Two main parts were presented in this manuscript. The first part reported an optimized purification process by response surface methodology to obtain high-purity neferine. The second part reported anti-metastasis mechanism of neferine for its anti-cancer activity on HepG2 cells. Both parts are not suitable for publication in their current contents, and should be substantially improved before reconsideration.
In the first part, the authors claimed to design a new protocol for purification of neferine more efficiently and environmentally friendly. However, there was no comparison between the current protocol and the known ones as neferine is commercially available and isolation protocols have been reported previously; even a patent (Method for extracting liensinine, isoliensinine and neferine from lotus plumule) has been documented though expired. Therefore, the authors have to compare with other protocols and pinpoint the advancement and advantage of their currently developed protocol.
In the second part, anticancer effects of neferine have been reported as summarized in a review article (Pharmacological benefits of neferine - A comprehensive review. Marthandam Asokan et al, Life Sci. 2018;199:60-70). Therefore, the authors have pointed out the new information provided by their work instead of showing the anti-cancer activity of neferine simply.
Reviewer 2 Report
1. Authentication of plant and detailed isolation need to be explained.
2. Grammatical errors are seen in several places.
3. Thoroughly check the references.
4. Therefore, authors are asked to resubmit after the modifications as per the suggestions.
Grammatical errors are seen in many places.
Botanical name of the plant species name should start with small letter.
Round 2
Reviewer 1 Report
Two main concerns I raised in the original version were adequately clarified in the Discussion of the revised version. Thus, the manuscript is recommended for publication.